# Key topics in pandemic health risk communication: A qualitative study of expert opinions and knowledge

**Siv Hilde Berg**[ID]*, **Marie Therese Shortt, Jo Røislien**[ID]**, Daniel Adrian Lungu**[ID], **Henriette Thune, Siri Wiig**

Faculty of Health Sciences, Department of Quality and Health Technology, Centre for Resilience in Healthcare, University of Stavanger, Stavanger, Norway

* Siv.h.berg@uis.no

## Abstract

### Background

Science communication can provide people with more accurate information on pandemic health risks by translating complex scientific topics into language that helps people make more informed choices on how to protect themselves and others. During pandemics, experts in medicine, science, public health, and communication are important sources of knowledge for science communication. This study uses the COVID-19 pandemic to explore these experts' opinions and knowledge of what to communicate to the public during a pandemic. The research question is: What are the key topics to communicate to the public about health risks during a pandemic?

### Method

We purposively sampled 13 experts in medicine, science, public health, and communication for individual interviews, with a range of different types of knowledge of COVID-19 risk and communication at the national, regional and hospital levels in Norway. The interview transcripts were coded and analysed inductively in a qualitative thematic analysis.

### Results

The study's findings emphasise three central topics pertaining to communication about pandemic health risk during the first year of the COVID-19 pandemic in Norway: 1) how the virus enters the human body and generates disease; 2) how to protect oneself and others from being infected; and 3) pandemic health risk for the individual and the society.

### Conclusion

The key topics emerging from the expert interviews relate to concepts originating from multiple disciplinary fields, and can inform frameworks for interprofessional communication about health risks during a pandemic. The study highlights the complexity of communicating pandemic messages, due to scientific uncertainty, fear of risk amplification, and

public repository. Fig 1 shows how codes are represented by the participants (coding three). The interview guide is found in S1 File. Data are available from the SHARE Center at the University of Stavanger, Norway (contact via share@uis.no) for researchers who meet the criteria for access to confidential data.

**Funding:** The COVID communication: Fighting a pandemic through translating science (COVCOM) project has received funding from the Trond Mohn Foundation under grant agreement number TMS2020TMT10 and the University of Stavanger. The funders had no role in study design, data collection and analysis, decision to publish, or preparation of the manuscript.

**Competing interests:** The authors have declared that no competing interests exist.

heterogeneity in public health and scientific literacy. The study contributes with insight into the complex communication processes of pandemic health risk communication.

# 1 Introduction

In a pandemic, a virus capable of infecting humans spreads and crosses international borders, usually affecting a large number of people [1]. Examples of pandemics include the "Spanish Flu" (1918–1919), "HIV/AIDS" pandemic (1966-), "Swine Flu" (2009–2010), and the "COVID-19" pandemic (2019-).

A pandemic associated with an unknown virus brings uncertainty that challenges politicians' and policy makers' decision making. Scientific uncertainty gives rise to a difficult balancing act in public communication: being transparent about the limits of human knowledge, yet still communicating clear messages [2]. The challenge is reinforced by poor public health literacy, which impedes people's ability to understand the rationale for governmental recommendations and to anticipate the consequences of their health choices [3]. To strengthen peoples' scientific-and health literacy, science communication should convey accurate information on health risks in a language that enables people to make independent and informed choices that will protect both themselves and others [4].

Scientific uncertainty, combined with the difficulty of delivering timely but complex messages to the population in a competitive media market, highlights the importance of health risk communication being not only correct and trustworthy, but also engaging. What to communicate to the public about staying safe during a pandemic–and how to communicate it–thus needs to be done with tremendous care [5].

Experts in medicine, public health, science and communication are important sources of knowledge in science communication during public health crisis [4,6]. This study uses the COVID-19 pandemic to explore experts' opinions and knowledge of what to communicate to the public during a pandemic. The research question is: What are the key topics to communicate to the public about health risks during a pandemic?

## 1.1 Health risk communication

Public communication during health emergencies aims to improve health outcomes by influencing, engaging, and reaching out to different at-risk audiences [7,8]. Public communication about a pandemic is, as with all communication, a multifaceted field in which the contents of the message, the messenger and the qualities of the audience attributes affect the reception of the message itself [8–11]. In this section, we present four topics which are relevant to this study's findings regarding the experts' communication of key topics: 1) trust and communicating uncertainty; 2) tradeoffs in decision making; 3) risk amplification; and 4) public engagement.

**1.1.1 Trust and communicating uncertainty.** Communicating honestly about uncertainty can build public trust and increase the legitimacy and credibility of the decision making process [12]. Public trust in the messengers of pandemic health risk information is associated with their willingness to comply with measures to prevent infection [13,14]. The World Health Organisation (WHO) advises health authorities to be candid about risk, events and interventions and to be forthcoming about what is known and what is not [7]. However, the different types of uncertainty that have been communicated have different effects on trust. Gustafson and Rice [15] reviewed the experimental literature and found that communicating

disagreements or conflict in science (*consensus uncertainty)* were most often associated with reduced credibility. Communicating *technical uncertainty* (e.g., probabilities) had positive or no effect on perceived credibility. The effect was mixed in relation to communicating *scientific uncertainty*, which had both positive and negative effects on perceived credibility.

**1.1.2 Tradeoffs in decision making.** Several safety science theorists have outlined *trade-offs* as an inevitable balancing act among all kinds of pressures and safety in hazardous systems [16–18], including healthcare [19]. Tradeoffs involve ambiguity over what constitutes the right answer. According to Gregory et al. [20] tradeoffs are usually difficult to communicate. Changes in one endpoint are often achieved at the expense of other goals, and the consequences of tradeoffs are often unclear to the decision makers. According to Gregory et al. [20] tradeoffs reflect issues that science cannot resolve; they depend on values in making decisions about risk. Little is known about the public behavioral effect of communicating tradeoffs on decision making in pandemics, yet Gregory et al. [20] and Norheim et al. [21] argued that tradeoffs should be transparent to foster trust.

**1.1.3 Risk amplification.** The Social Amplification of Risk Framework (SARF) suggests that most health risk messages are sent through transmitters (e.g. media, messengers) which alter the original message by intensifying or attenuating some signals before passing them on [10]. SARF proposes that risk deemed less urgent by experts may be *amplified* by other transmitters; however, risk that experts consider less urgent may be *attenuated*. According to SARF, silence from experts and decision makers may breed fear and suspicion among those at risk and lead to risk amplification. Nevertheless, a moderate level of fear can mitigate the risk of a pandemic by encouraging behaviour that protects public health [22]. To minimise risk amplification, health risk communicators should avoid sensationalism, speculation and disturbing images [23].

**1.1.4 Public engagement.** In the early 1980s, "lay" perspectives were perceived as fallible and subjective by communication experts and policy makers, and communication interventions aimed to align lay perspectives with those of the experts in a *one-way approach*. As the field of health and risk communication evolves, multi-way approaches to communication emphasising *public engagement* have largely replaced the one-way approach to communication [9]. The WHO advises health authorities to involve the public in decision making to ensure that interventions are collaborative and contextually suitable, thereby constituting *a two-way approach* [7]. There is growing awareness of the benefits of including patients and the public in healthcare. However, involvement needs to be conducted at an even wider system level, e.g., involvement in the design of public health communication [24].

## 2. Method

The present study applies a qualitative design using individual interviews with Norwegian experts in medicine, science, public health, and communication guided by the mental models' approach to risk communication [4,25]. The mental models' approach to risk communication involves both experts' knowledge and opinions of what people should know to make informed decisions about a topic (normative research) and the ability of the intended audience (descriptive research) to create tailored risk communication interventions (prescriptive research) [4]. This study reports on the normative research phase [26]. See Berg et al. [27] for results from the audience analysis, and Røislien at al. [28] for the design of the intervention.

Expert opinions are valuable components of communication designs when the scientific evidence is missing, is inconsistent or incomplete, as is often the case with new viruses such as the coronavirus (SARS-COV-2) [25]. Prior research of expert mental models' have used health scientists' and medical practitioners' knowledge of infectious diseases to design health risk

communications [29,30]. To our knowledge, no mental models' studies of infectious diseases have so far included the knowledge of communication experts.

## 2.1 Participants and sampling strategy

Communicating key topics during a pandemic is a dynamic process in which a range of inter-professional issues of pandemic health risk are attributed different temporal weights. Infection control was the Norwegian government's and policy makers' main focus in the first wave of the pandemic, while mental health and public health became increasingly important as the pandemic evolved [31]. Communication science is an important field of knowledge to pandemic responses [7], however it is not necessarily emphasised as part of the public health risk communication field. We therefore sought informants with expertise in medicine, science, public health, and communication to cover the variety of expertise relevant to pandemic risk communication.

We purposively sampled 13 experts in medicine, science, public health, and communication for individual interviews, all with extensive knowledge of COVID-19 and risk communication at the hospital, regional and national levels in Norway. Participants who met the inclusion criteria were identified and recruited by invitation from the research group. One participant dropped out of the study due to time issues.

## 2.2 Interviews

A semi-structured interview guide was developed to explore key topics related to communication about COVID-19 virus transmission and exposure, mitigation management, and potential effects of infection (S1 File). The interview guide also explored key topics related to pandemic health risk generally and those in each expert's specialty. The interview guide was pilot tested with participant number one. The interviewer had no prior established relationships with the participants, and the participants were informed about the reasons for this research (exploring key topics in pandemic communication and their experiences with communication). Due to national COVID-19 measures to keep physical distance, the participants were interviewed individually by SHB using a video conferencing software program (Zoom). Mean duration of interviews was 59 minutes (range 46–80 min). Interviews were audio recorded and transcribed. Ethical approval was obtained from the Norwegian Centre for Research Data (Ref nr. 583192). All participants gave written, voluntary and informed consent. The interviews were conducted between February and April 2021.

## 2.3 Context

The data material in this study presents important topics for a first-year pandemic response. At the time of data collection, individuals in Norway with the highest risk of severe consequences of COVID-19 were being vaccinated. The Alpha variant of the coronavirus (B.1.1.7) had started to spread in the Norwegian population and led to the third wave of COVID-19.

In the first year of the pandemic, the number of positive cases and deaths per million due to COVID-19 were much lower in Norway than in other European countries [32]. Since the first wave of COVID-19 struck Norway in March-April 2020, the government strategy has been to limit the spread of infection by using testing, isolation, contact tracing, and quarantine, and by occasionally closing schools and kindergartens, and cancelling cultural events [33]. The goal has been to control the spread of infection so that the infection is manageable and does not exceed the capacity of the health and care service and the municipal health service [34].

Norway has a strong public healthcare system; most hospitals are funded and owned by the state. The welfare state is well-developed and founded on the principles of universal access,

decentralisation and free choice of provider [35]. Compared to other OECD countries, Norway is a high-trust society [36]; in a recent study 96% of the participants reported that they trusted the health authorities in general [37]. Trust in the health authorities has remained high and stable during the pandemic; between 80 and 90% of the population expressed a high level of trust in the health authorities [31], and especially in the Norwegian Institute of Public Health (NIPH) [38]. Norway has a population of 5.4 million. Although the country has the third lowest population density in Europe, at the beginning of the COVID-19 outbreak, Norway was second only to Italy in the number of confirmed cases per capita. However, by April 2020 Norway had been able to "flatten the curve" and ease the competition for hospital beds [38]. A strategy used by the Norwegian government during the first pandemic response was to appeal to people's sense of solidarity by using the term *dugnad*, an old cultural practice that describes volunteer activities that will benefit the community [39].

## 2.4 Analysis

The expert interview transcripts were coded and analysed inductively in a qualitative thematic analysis [40]. The unit of analysis was "what people should know", and the analysis followed the five phases of systematic thematic analysis as outlined by Braun and Clark [40]. To become familiar with the data, all transcripts were read by SHB and MT (phase 1), who discussed first impressions of the material with SW. SHB inductively coded data across the entire data set related to the meaning units (phase 2). The initial codes were extracted to tables, which were used to search for meaningful patterns, themes, and levels of themes. All co-authors reviewed the initial organisation of codes. The codes were then compiled under 36 key messages, which were then abstracted into seven sub-themes and three themes (phase 3). In phase 4, a table with thematically organised codes was made to validate the themes in relation to the codes and the entire data set. The themes were refined and renamed to generate clear terms for each theme (phase 5). The fifth and final phase consisted of writing out the results with codes and extract examples, which all co-authors read and discussed.

Thematic mapping is used in thematic analysis to find and display relationships between codes and themes [40] and influence diagrams are often used to visualise experts mental models [25,26]. However, these visualisation techniques are one-dimensional and do not allow for multi-layered information to be displayed. A *bubble chart* was thus created; a visualisation method developed by Shortt et al. [41] for visualising qualitative interview data, inspired by thematic mapping and influence diagrams [25,40]. The bubble chart displays topics emphasised by experts in medicine, science, public health, and communication, with colour coding that enables visual comparison of topics emphasised by the different experts. Topics with disagreement whether they should be communicated to the public or due to being perceived as not relevant, to complicated or too uncertain to communicate to the public, are highlighted.

## 3 Results

Participant characteristics and the participants' field of expertise, professional role during the COVID-19 pandemic and in public communication are available in Table 1.

The first impression of the data material was linked to how seriously the medical and science experts took their responsibility for communicating pandemic risk to the public, yet they emphasised not being trained in public communication. The medical and science experts were reflexive in crafting each message to each audience, and in maintaining their ethics in public communication. They expressed personal uncertainty in their role as communicators to the public. Although the communication experts had been trained, none of them had ever had to issue communications about such an ambiguous situation, especially one of the scale of the

**Table 1. Sample characteristics.**

| | Field of expertise and sample category | Professional role | Public communication role related to COVID-19 | Gender |
|---|---|---|---|---|
| 1 | Public health management, nurse (Public health) | Head of department for community healthcare service. Infection tracing and training of healthcare professionals | Advisor of municipal communication to the public | F |
| 2 | Specialist in community medicine (medicine) | Municipal chief physician and head of COVID-19 emergency room. Infection prevention at individual and group level | Municipal communication to the public | M |
| 3 | Public emergency and risk management (public health) | Head of municipal emergency management and community risk communication | Municipal communication to the public | M |
| 4 | Marketing (communication) | Audience analysis, targeted communication to adolescents and young adults | Advisor of municipal communication to the public | F |
| 5 | Clinical psychologist, PhD in psychology, journalism (public health) | Psychologist in primary mental healthcare | Municipal communication to the public | F |
| 6 | Communication and journalism (communication) | Audience analysis. Targeted communication related to motivation and mental health | Advisor of municipal communication to the public | F |
| 7 | Professor and infection medicine specialist (medicine) | Head of infection medicine in a Regional Hospital. Infection prevention at individual and group level | Communication to the public at regional level | M |
| 8 | Infection medicine specialist (medicine) | Infection prevention at hospital level. Microbiology, virus transmission and treatment of COVID patients | Science dissemination in medical journals | F |
| 9 | Infection medicine specialist (medicine) | Head of infection medicine in a Regional Hospital. Infection prevention at individual and group level | Communication to the public at regional level | M |
| 10 | Risk communication and ethnography (communication) | Director of nationwide health risk communication | Strategy level, nationwide health risk communication | F |
| 11 | Professor of statistics (Science) | Biostatistician. Mathematical modelling of the R-value and the spread of COVID-19. | Communication to the public at national level | M |
| 12 | Professor of health communication and cell-biologist (Science) | Health communication researcher. Health literacy in the general public | Science dissemination in national media | M |
| 13 | PhD and infection medicine specialist (Science) | Infection medicine researcher. Virus transmission, immunity and antibodies related to SARS-COV-2 | Communication in national media | F |

COVID-19 pandemic. All of the experts had to learn about this type of communication during a pandemic by doing it, despite scientific uncertainty and constantly changing evidence.

The analysis of expert opinions and knowledge identified 36 key messages in pandemic health risk communication to the public which were organised into seven sub-themes as part of three main themes: how the virus enters the human body and produces disease; how to protect oneself and others from infection; and pandemic health risk for the individual and the society (Table 2).

How each of the expert groups emphasised each of the themes and sub-themes is visualised in Figs 1–3. An animated version of the bubble chart reveals the multi-layer organisation of the themes and sub-themes according to the various expert groups (S2 File).

**Table 2. Key topics to communicate to the public related to pandemic risk.**

| Themes | How the virus gets into the human body and generates disease | How to protect oneself and others from being infected | Pandemic health risk for the individual and the society |
|---|---|---|---|
| Sub-themes | Modes of virus transmission | Infection prevention at the individual level | Solidarity |
| | Virus and immunity | Infection prevention at group level | Control of the spread |
| | | | Risk tradeoffs |

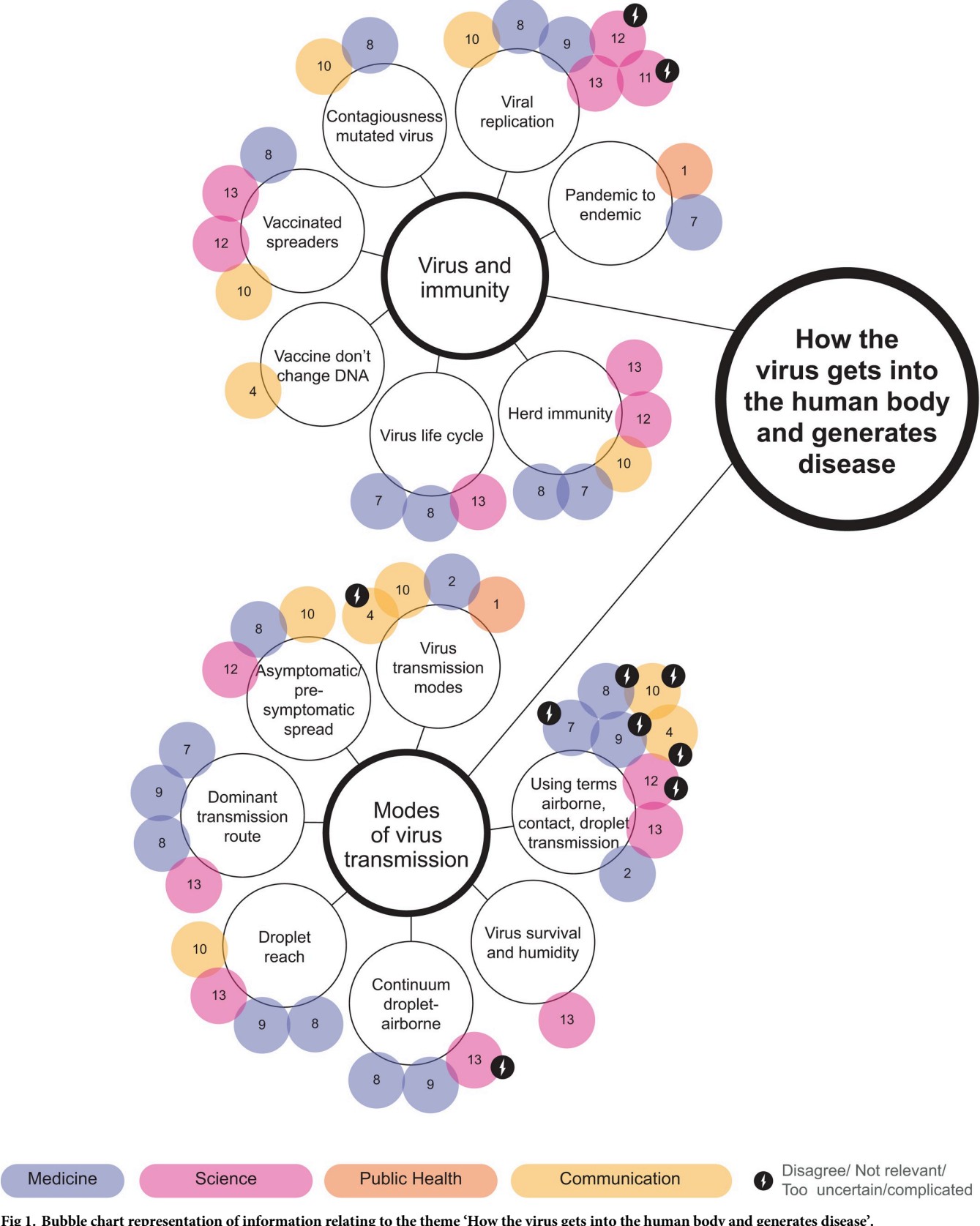

**Fig 1. Bubble chart representation of information relating to the theme 'How the virus gets into the human body and generates disease'.**

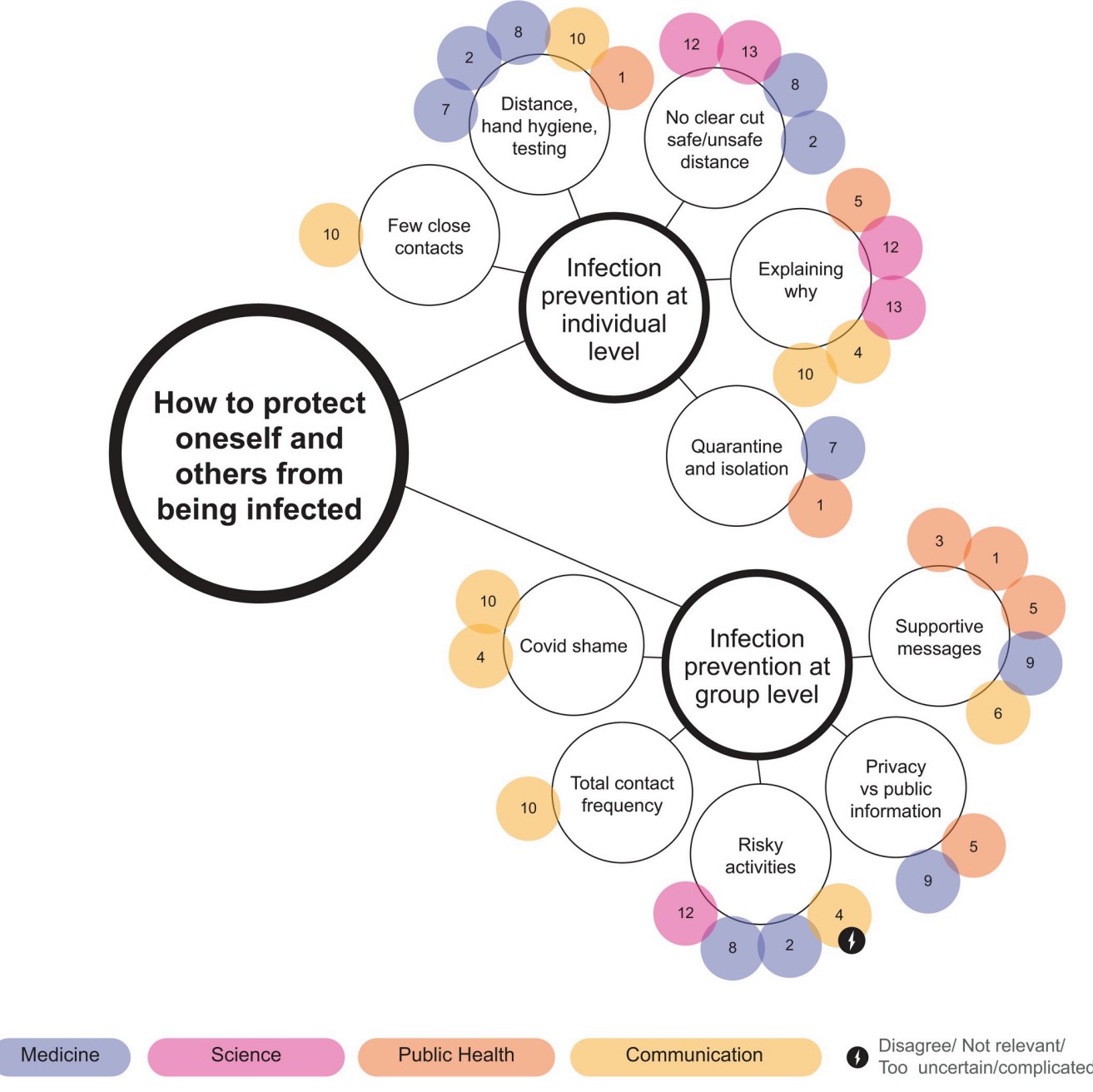

**Fig 2. Bubble chart representation of information from interviews relating to the theme 'How to protect oneself and others from being infected'.**

### 3.1 How the virus gets into the human body and generates disease

The theme "how the virus gets into the body and generates disease" has two sub-themes: modes of virus transmission, and virus and immunity (Fig 1).

**3.1.1 Modes of virus transmission.** The experts' emphasis on modes of virus transmission is related to a potential increase in people's comprehension of why they should comply with recommendations to protect themselves from COVID-19. One necessary message to communicate was how droplets containing the virus could reach people within a metre of the

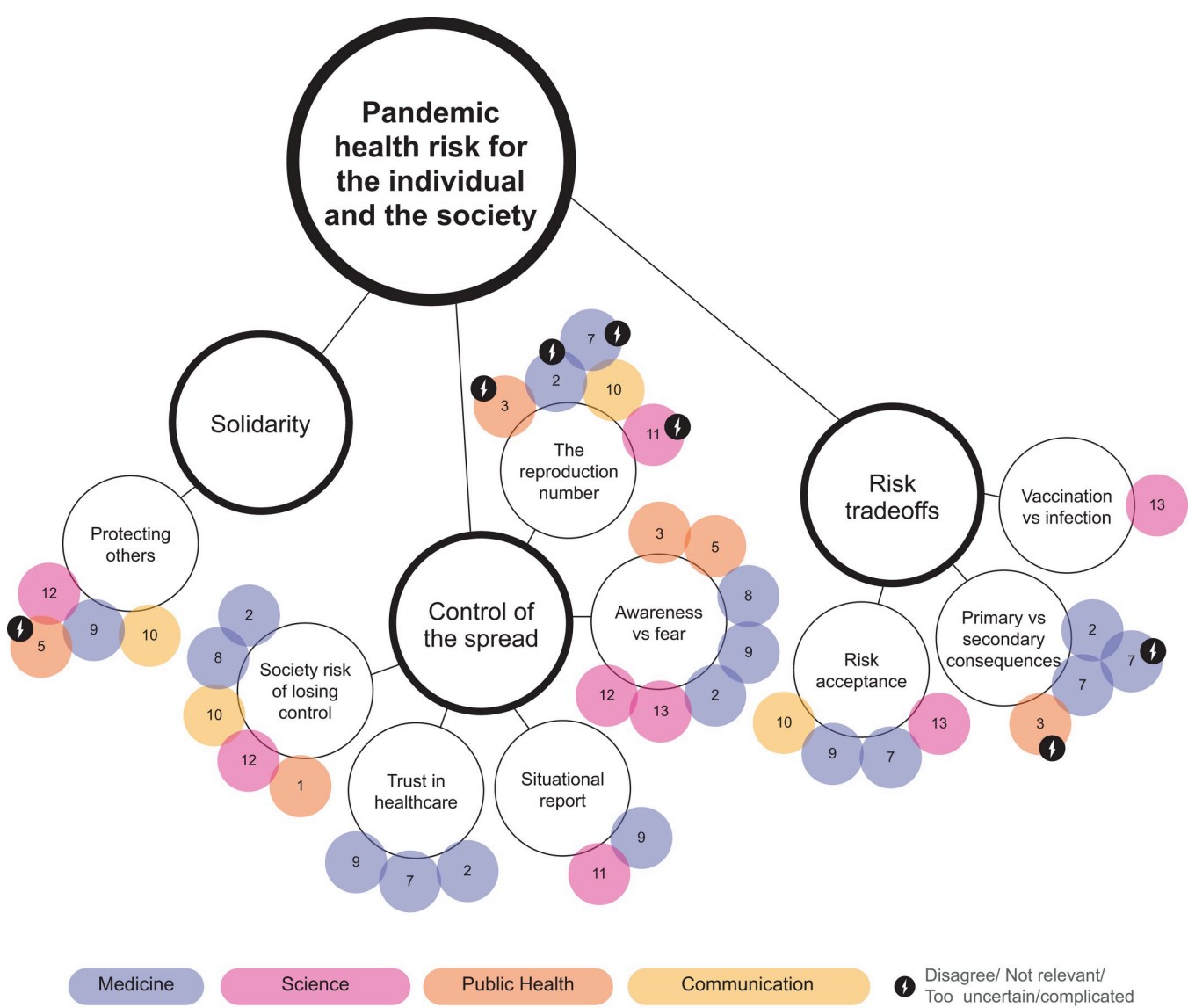

**Fig 3. Bubble chart representation of information from interviews relating to the theme 'Pandemic health risk for the individual and the society'.**

person emitting the droplets. This message helps to reinforce the need to keep physical distance.

Experts from several fields insisted that the public needed to comprehend the concepts of *incubation period* and *asymptomatic transmission*, which they expressed as: "You can be contagious even if you don't have symptoms. Non-symptomatic people can carry the disease without being aware of it. You cannot tell if a person is infected or not, so everyone needs to keep distance".

At the time of data collection there was scientific uncertainty about the modes of virus transmission of the coronavirus. Experts in medicine highlighted that it was important to address this uncertainty by communicating the *most dominant route of transmission* to help people distinguish *possible* from the *most probable* way of transmission.

Some of the experts used the terms *contact/surface transmission*, *droplet transmission* and *airborne transmission* when communicating with the public. The experts agreed that there was

not a clear distinction but rather a continuum between droplet and airborne transmission. Although one expert in medicine believed it was important to illustrate a continuum of droplets with different sizes and reach, another medicine expert communicated how temperature and humidity affected how long the droplets stayed in the air and thus increased the probability of virus survival, and therefore the chance of being infected.

Contact transmission was found to be difficult to explain, due to the scientific uncertainty, as people who received the message interpreted the same information in different ways; as deadly with the need to go to extremes or as a non-relevant risk.

Some of the medicine experts did not communicate airborne transmission to the public. They considered that communicating the topic could elicit excessive fear due to the scientific uncertainty related to airborne transmission and SARS-COV-2, and due to the high level of scientific/health literacy needed to comprehend how aerosols could stay in the air. The science expert described how creating mental images of the airspace around people by using analogies to water have been used successfully as a pedagogic tool:

*People have rather vague ideas about the medium of air, and what it means to breathe at a distance of one metre. It spreads in the same way, as if you dip a teabag in a cup of water. Because there is no hole here, it's compact matter. . .. These particles are carried through matter in the same way as if you had dropped a small wooden plank on the water surface (No. 12)*

Experts who did not use the terms *droplets*, *airborne* and *contact/surface transmission*, believed they were too technical and ambiguous. Instead, they prioritised the conveying of clear messages; how people could protect themselves by washing their hands and keeping their distance, as exemplified by a communication expert for nationwide health risk communication:

*We have not been completely oblique about how the virus infects, and new virus variants change how the virus behaves. . .We have not used the terms airborne, droplet and contact transmission so often. We have been more concerned with explaining what the best precautions to take to avoid being infected by the virus are. . .We have noticed that there is a great demand among the general public for information about the mode of transmission, but people are just as concerned about how to handle the infection. They want to know what to do (No. 10).*

One of the communication experts pointed out that modes of virus transmission were not mentioned in their target group analysis of young people's concerns and values, thus the topic was not included in their targeted communication to adolescents.

**3.1.2 Virus and immunity.** The medicine experts believed that it was important to explain how a *virus attaches itself to the cells* and creates disease. Such mechanisms were important to increase peoples understanding of variant virus' changes in transmission. The variants of the virus had different causes for their increased contagiousness; for example, a medicine expert explained how a virus which can be in the upper respiratory tract one to two days before the disease breaks out is extremely contagious. A communication expert believed that explanations of virus autonomy and multiplication were too complicated for most people to understand. The communication expert for nationwide health risk communication noted that they had included this information when communicating to the public, despite its complexity:

*We have tried to explain that here is a part of this virus, this spike that attached better [SARS-COV-2 Spike Protein]. It is complex information, but we have really thought that it is*

*important to try to give it anyway, so that people understand why we still ask them to keep distance, and maybe even keep more distance now, because the contagiousness in the virus variations is different to the first virus variant that we are familiar with (No. 10)*

Medicine experts experienced that the public tended to perceive *mutations of viruses* as dangerous, and considered it important to communicate the message that mutations are a natural part of the virus's life cycle. The virus spreads by becoming more contagious without killing its host. Thus, by mutating the virus becomes less lethal.

Experts from diverse fields considered it important to inform about the effects of vaccination on immunity and how vaccination affected the individual and the society. At the individual level, it was important to convey the message that "you can still be infected and infect others even if you are vaccinated, however the vaccine may limit the lethality of infection". At the society level, reducing the risk of infection through *herd immunity* through mass vaccination was a relevant concept to communicate.

## 3.2 How to protect oneself and others from being infected

The theme "how to protect oneself and others from being infected" pertained to two sub-themes: infection prevention at the individual level, and at the group level (Fig 2).

**3.2.1 Infection prevention at the individual level.** Experts from various fields agreed that the basic infection prevention measures were the most important to communicate to the public: keep distance; wash your hands; stay at home and get tested if you show symptoms; limit your contacts; and what quarantine and isolation mean. The communication expert for nationwide health risk communication explained the importance of conveying core messages with clear instructions:

*We have observed a rather surprising desire for clear messages. . .There is a much greater willingness to be told what to do in Norway than I was prepared for. . . It's very special, and a little different from the risk communication theory that state that you should describe the risk and let people make their own decisions (No. 10).*

However, the rules were difficult to enforce if people did not understand the underlying principles. Many of the experts insisted that the rationales for infection prevention measures were important to explain to the public. Explaining *why* people had to follow the rules, and being open about the evidence and the pandemic decision making, were considered important in the creation of trust.

The science and medicine experts considered it important to teach the public about probabilities, and to convey the message that "there are no clear boundaries between safe and unsafe distance, the larger distance you keep, the less probability you have for being exposed to infection". According to the science expert:

*It is important to emphasise that when you set a two-metre limit, it is not because there is a black-and-white limit that if you are 2 metres and 10 cm from someone, you are safe, and if you are 1 metre and 90 cm, then you are not. That's silly. So, you have to trust people to use their common sense and understand that it is a question of probabilities in all of the rules and guidelines that exist (No.13).*

**3.2.2 Infection prevention at group level.** While the scientific evidence for infection prevention was more definitive at the individual level, higher scientific uncertainty was related to

infection prevention at the group level, which often led to public debates in the media over whether or not the evidence supported lockdowns of specific activities. The medicine experts emphasised the need to communicate the risk of being physically close to others and engaging in activities that increased the expulsion of droplets containing the virus (e.g., singing, going to pubs and gyms) to increase people's understanding of why those activities had been stopped. As the nationwide health risk communication expert emphasised:

> *It is not about whether it is correct to close that particular school or shopping centre or not, it's about reducing the total contact frequency among individuals, and these choices will be implemented with a high level of uncertainty regarding their efficiency (No. 10).*

To reduce the stigma of contracting COVID-19, the communication expert for nationwide health risk communication emphasised that "this virus does not differentiate between people". However, some of the medicine experts at community and regional hospitals reported feeling trapped between safeguarding individuals' right to privacy and satisfying the media's need for information.

Experts emphasised that conveying a message of hope could help people remain motivated despite of social distancing. The medical experts emphasised supportive messages and acknowledging people's compliance. A communication expert described tailoring different messages to different populations in the municipality by showing them activities they could do during lockdown:

> *We identified that people wanted the municipality to stop telling them what they could not do, but rather what they could do. People can be outdoors even if we are on red level, and we have to think about measures to support, that involves being outdoor (No. 6).*

### 3.3 Pandemic health risk for the individual and the society

The theme "pandemic health risk for the individual and the society" was related to three sub-themes: solidarity, control of the spread, and risk-tradeoffs (Fig 3).

**3.3.1 Solidarity.**   Experts explained the importance of informing the public of the collective moral responsibility to protect others, appealing to Norwegians' sense of solidarity. As the communication expert at the national level explained:

> *Initially we needed to bring the entire population to the same problem definition, because we need to share the problem. We explained that it was most dangerous for the oldest and groups with chronic diseases. . .This means that approximately 70 percent of the population should be careful to protect the remaining population, and that as an individual you get some tasks to do to protect someone else than yourself (No. 10).*

However, appealing to solidarity was considered challenging, because the messages need to be understood by the whole population, including people who are not in a life or work situation where they can participate in the collective effort by keeping physical distance, and people who come from cultural backgrounds that have a different understanding and valuation of collective moral responsibility.

**3.3.2 Control of the spread.**   Experts from different fields stressed the need to communicate the risk of losing control of the spread. They stated that "while the pandemic risk is low for many individuals, it is high for society". To express the risk to society, the experts reiterated

the potential consequences and future scenarios, conveyed as: "The virus is highly contagious. Losing control of the spread of the virus, will lead to healthcare services becoming overloaded which makes the pandemic deadlier".

The media often used the reproduction number (R-number) to describe the status of the spread. According to the science expert in biostatistics:

*The reproduction number says something about the growth of the disease, so it is the speed with which the disease will grow, the number of infected people will grow. And if this number is above 1, then the epidemic is exploding, if it is below 1, it is under control and will disappear. So, it is a fundamental number that describes the current status, where are we now (No. 11).*

However, many of the experts decided against mentioning the R-number in their public messaging, because of the risk of misunderstanding it. The R-number was considered too complicated to explain to the public due to the need to understand uncertainties, simplified assumptions, and exponential growth. Many of the experts believed that the R-number could result in a misleading picture of the pandemic.

The experts stated that explanations of the risk of losing control should be balanced with information empowering the population to control the spread. Descriptions of realistic risk scenarios needed to be balanced with messages of hope. It was just as important to convey the severity of the risk as it was to avoid creating panic. As the public health expert in community psychology stated:

*If too much fear is triggered, I think some people will defend themselves and refuse to deal with it. You should neither make the message so dangerous that it becomes overwhelming, neither so harmless that people think it does not matter what they do. I think you cannot undercommunicate the consequences. You have to get a balance there (No. 5).*

Lastly, messages implying the risk of losing control needed to be balanced with messages strengthening people's sense of control. As such, the medicine experts did not merely reveal the status of infected and hospitalised people, but also that "we have control" to ensure the public's trust in healthcare delivery. The experts believed in creating situational awareness while offering specific advice on how to minimise their risk.

**3.3.3 Risk tradeoffs.**   Some experts believed that giving people insight into their dilemmas could improve the public's understanding of management strategies for the pandemic.

The experts emphasised the significance of informing the public of the risk tradeoffs between primary and secondary consequences of the pandemic". However, their perspectives on risk differed. The medical experts at hospital and regional level had low risk acceptance for the negative consequences of lockdown and isolation (e.g., increased mental illness among children and adolescents). Thus, they wanted all measures to be proportionate to the infection risk. Nevertheless, the public health expert in community emergency management had low risk acceptance for primary consequences of the pandemic, and thus communicated the importance of preventing the spread of the virus by implementing measures as early as possible:

*It's very difficult to use strict and invasive measures to prevent infection. You have to wait until the infection has spread. Different professional experts, health and emergency preparedness disagree. We [community emergency preparedness] believe it is not about spreading fear. It is about having a proactive approach around a potential hazardous situation in the future.*

*It is difficult to communicate, because our message is often shot down by the central health authorities (No. 3).*

The science and medicine experts contended that the risk tradeoff between being infected versus being vaccinated needed to be communicated. This tradeoff was difficult to express, especially at the individual level:

*To communicate that vaccines are used when a virus is so dangerous that it is worth taking the risk of side-effects to avoid being exposed to the infection is challenging. Where is that threshold? There is no final decision on that. . . When is the intersection where your individual risk should actually take precedence over the best interest of society and herd immunity? It's twisted. I try to communicate the dilemma. I do not have a definitive answer, because there is no definitive answer (No. 13).*

## 4. Discussion

This study uses COVID-19 as a case to explore medicine, science, public health and communication expert' opinions and knowledge of topics to communicate to the public in relation to pandemic health risks. The discussion of the findings is related to the interdisciplinarity of the emerging key topics, communicating uncertainty and messages which may amplify risk, and the role of public engagement along with expertise knowledge in designing messages.

### 4.1 Interdisciplinarity

This study has identified three key topics of scientific knowledge in pandemic health risk communication. The experts focused on viral replication and modes of virus transmission, protection from the disease, and the health risk for individuals and the society. These key topics relate to concepts which originate from different fields and disciplines, including infectious diseases (e.g., virus transmission, immunity), psychology (e.g., fear, hope, motivation), philosophy (e.g., collective moral responsibly), statistics (e.g., R-number), risk communication (e.g., situational awareness, risk tradeoffs), health communication (e.g., health literacy), pedagogy (e.g., visual analogies), and marketing (e.g., targeted communication).

Furthermore, the different experts had different priorities in terms of what messages to convey. This poses a risk for contradictory communication. The multi-disciplinarity of the key topics in this study, and the different views on key topics to communicate, demonstrate that messages should not be developed from one scientific field only, but should rather rely on interdisciplinarity to ensure coherence, clarity, and engagement. For example, to communicate a complex topic such as the R-value requires experts to work across the boundaries of different disciplines, e.g., statistics, medicine, pedagogy, and marketing, which are all needed to together create and tailor the communication intervention.

### 4.2 Communicating uncertainty and messages which may amplify risk

In this study, the omission of talking about the R-number and airborne transmission in public communication were related to the experts' belief that raising these topics characterised by scientific uncertainty could amplify the risk. However, under those circumstances people may consult other sources of information, which may lead to misinformation and consequently risk amplification [42].

Nevertheless, evidence does not support the communication of uncertainty on all occasions. The communication of uncertainty can have a range of outcomes. According to

Gustafson and Rices' [15] review of the experimental literature on different types of trust, explaining the confidence interval related to the R-number and explaining probabilities related to infection prevention measures may increase public trust, as these are related to communicating technical uncertainty [15]. Nonetheless, communicating uncertainty about airborne transmission at the early phase of the pandemic may undermine public trust, as this type of uncertainty is related to scientific uncertainty, and lack of experts' consensus [15].

The experts emphasised being open and transparent and communicating tradeoffs in crisis management as vital to the creation of trust and strengthening people's comprehension of the rationale behind infection prevention measures. Trust is essential during rapidly evolving events characterised by scientific uncertainty [9,13,14]. The experts reflect that the pandemic measures have both intended and unintended consequences, forcing decision makers to trade short-term policy goals for unintended long-term effects [43]. Recent studies have shown that quarantine, staying home and closing schools have implications for unemployment [44], mental health issues [45,46], interrupted learning [47] and domestic violence [48]. Countries strive to balance safety and health with economic security and personal freedom [49]. Norheim et al. [21] argue that by explaining these dilemmas, people learn about the complexity of the decision making process, and strengthen social trust. Gregory et al. [20] argue that experts and authorities should be transparent and acknowledge value-based judgements in environmental risk management, rather than pretend that the decision making is objective and non-value-based. Consistent with the literature, this study identifies risk tradeoffs in pandemic health risk communication as a topic to communicate. Tradeoffs can be conveyed by the messenger in their messages through transparency about what is at stake and how the interests of different parties are assessed and weighted. However, through interviewing participants at municipal, regional and national level, this study found that experts experienced different risk trade-offs related to primary and secondary consequences of the pandemic. Such risk tradeoffs, may depend not only on their area of expertise, but also on their responsibility in the crisis management and their perspectives on risk and their proximity to the hazard. Being transparent about such tradeoffs may introduce the risk of fragmented messages due to experts' different perceptions of risk. When experts disagree, dissemination of risk tradeoffs may unintentionally compromise peoples' trust by amplifying people's perception of risk [15]. Thus, communicating tradeoffs needs to be addressed differently depending on how each risk is perceived, contextualised and communicated by others in the crisis management system.

Furthermore, in this study, medical experts emphasised the balance between providing people with facts and instilling fear. The experts protected this balance by suggesting future scenarios and the consequences of uncontrolled virus spread with the aim of creating a realistic picture of the threat while avoiding war-framing and exaggerations. Both social amplification and attenuation of risk undermine the effectiveness of risk communication [10]. A moderate level of fear is needed to take the risk seriously and act on it [22]. Consistent with the literature, this study identifies the balance between fear and efficacy as vital when communicating control of the spread [10,50]. By doing so, health risk communicators may provide the public with necessary information, without attenuation or amplification of the risk in correspondence with the SARF [42].

## 4.3 Public engagement

This study indicates that experts across professional fields are taking the audience's knowledge (or lack of) and concerns into account. The medical and science experts are mindful of relying clear messages, and of simplifying complex scientific topics and principles. However, science, public health and medical experts have different preferences from the communication experts

in terms of selecting the content of their messages. The medical and science experts, although acknowledging their audience, adopt a one-way approach that consists of identifying topics based on what they consider most important. The communication experts adopt a two-way approach that is intended to fill gaps in the public's knowledge.

There are limitations to both approaches. The recipients of communication must regard the message as relevant; this is a drawback of the one-way approach because without knowing their audience experts' communication may be too detailed and detached from public's values [9]. The drawback of communication interventions informed by audience analysis only is the omission of topics not emphasised by the target group. Pandemic health risk communication also needs to provide people with information that they do not realise they need to protect themselves and others. Communication created based on audience analysis, without expert opinions may fail to fill knowledge gaps [4].

There is no one-size-fits-all to the communication of health risk during a pandemic. There will always be unintended effects on some audience members and different considerations are taken at hospital, regional and national levels. Both audience analysis and interdisciplinary collaboration among experts in medicine, public health, science, and communication may facilitate the creation of engaging messages with intended outcomes [4].

## 4.4 Limitations

This study uses a normative expert study approach to collect multi-professional domain-specific knowledge [25]. Analytical generalisations made on a theoretical level can inform frameworks for inter-professional pandemic health risk communication and public health emergencies [51]. This study holds strong information power due to the quality of the interview dialogue [52], and the inclusion of participants with relevant experiences with the phenomenon under study [53]. However, the phenomenon of pandemic communication is ambiguous and complex, and a larger sample could have revealed an even larger breadth of pandemic communication topics. Future studies are needed to explore pandemic communication in other settings and phases of a pandemic with other fast spreading infectious diseases.

## 5. Conclusion

In this study we have explored experts' opinions and knowledge about health-related information topics that need to be shared with the public in a pandemic situation. By looking into the COVID-19 as a case, the experts in our sample revealed diverse key topics representing several disciplinary fields. The experts found it is fundamental to communicate the ways a virus enters the body and generates disease; the measures needed to protect oneself and others; and enabling a wider perspective on the individual and societal risks caused by the pandemic. This study demonstrates that pandemic health risk communication relies on interdisciplinary expertise across medicine, public health, medicine research, and communication. Experts emphasise not merely what to communicate but understand the message as meaning produced in the intersection of the messenger, the message attributes, and its audience. Thus, what to communicate is highly related to the complexity of communicating messages due to scientific uncertainty, fear of risk amplification, heterogeneity in public health and scientific literacy, among others. Therefore, the study contributes to the knowledge of complex communication processes of pandemic health risk communication.

The results from this study can be used to create communication interventions related to pandemic health risk, first of all by emphasising the need for a genuinely interdisciplinary approach to what needs to be communicated. The study contributes to the field of pandemic preparedness and informs effective communication interventions in future pandemic events.

Studies of intended audience members are required to identify what the public does and does not know, as advised by the mental models' approach to risk communication [4].

## Supporting information

**S1 File. Interview guide expert interviews.**
(DOCX)

**S2 File. Key topics animated bubble chart.** Animated bubble chart visualising concept mapping of expert opinions.
(MP4)

## Acknowledgments

We want to acknowledge the COVCOM user panel for providing feedback on our work in progress.

## Research team

This study was conducted by an interdisciplinary team of researchers in the Covid Communication project (COVCOM) lead by JR. SHB (PhD, Licenced Clinical Psychologist) is an associate professor with a background from psychology and safety science. She is an experienced interviewer. MT (PhD) is a postdoctoral researcher with a background from visual communication and media studies. JR (PhD) is a professor in medical statistics with a background from mass media communication and mathematics. DAL (PhD) is an associate professor with a background from experimental research and management studies. HT (PhD) is dean of research with a background from aesthetics, gender and intermedial studies. SW (PhD) is a professor with a background from safety science and risk communication.

## Author Contributions

**Conceptualization:** Siv Hilde Berg, Jo Røislien, Henriette Thune, Siri Wiig.

**Data curation:** Siv Hilde Berg.

**Formal analysis:** Siv Hilde Berg, Marie Therese Shortt, Siri Wiig.

**Funding acquisition:** Jo Røislien, Siri Wiig.

**Investigation:** Siv Hilde Berg.

**Methodology:** Siv Hilde Berg.

**Project administration:** Siv Hilde Berg.

**Resources:** Siv Hilde Berg, Marie Therese Shortt, Henriette Thune, Siri Wiig.

**Software:** Marie Therese Shortt.

**Validation:** Siv Hilde Berg, Marie Therese Shortt, Jo Røislien, Daniel Adrian Lungu, Henriette Thune, Siri Wiig.

**Visualization:** Marie Therese Shortt.

**Writing – original draft:** Siv Hilde Berg.

**Writing – review & editing:** Siv Hilde Berg, Marie Therese Shortt, Jo Røislien, Daniel Adrian Lungu, Henriette Thune, Siri Wiig.

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
