## [Decision Letter · Decision Letter 0]

9 May 2022

PONE-D-22-07588Key topics in pandemic health risk communication: A qualitative study of expert opinions and knowledgePLOS ONE

Dear Dr. Berg,

Thank you for submitting your manuscript to PLOS ONE. After careful consideration, we feel that it has merit but does not fully meet PLOS ONE’s publication criteria as it currently stands. Therefore, we invite you to submit a revised version of the manuscript that addresses the points raised during the review process.

 Please address all the comments and suggestions from the Reviewers.

We look forward to receiving your revised manuscript.

Kind regards,

Celia Andreu-Sánchez

Academic Editor

PLOS ONE

Journal Requirements:

Additional Editor Comments:

Dear Authors,

Based on reviewers' comments, my decision for your manuscript is minor revision. Find below the comments of Reviewers 1 and 2. Please, address all of their comments and suggestions.

Best regards,

Celia Andreu-Sánchez.

Reviewers' comments:

Reviewer's Responses to Questions

**Comments to the Author**

1. Is the manuscript technically sound, and do the data support the conclusions?

Reviewer #1: Yes

Reviewer #2: Yes

2. Has the statistical analysis been performed appropriately and rigorously? 

Reviewer #1: N/A

Reviewer #2: N/A

3. Have the authors made all data underlying the findings in their manuscript fully available?

Reviewer #1: No

Reviewer #2: No

4. Is the manuscript presented in an intelligible fashion and written in standard English?

Reviewer #1: Yes

Reviewer #2: Yes

5. Review Comments to the Author

Reviewer #1: This is a well written paper and I have only minor suggestions:

1. Please move duration of interviews to results section.

2. Please change 'UK variant' to 'alpha variant'

3. Please make the figures a bigger size so that they are readable.

4. Please consider shortening the results section.

Reviewer #2: The manuscript for this research article is well organized and clearly written. The overall study, construction, sample size, and design are very good.

The main claims are first, what exactly to communicate during a pandemic, in terms of risk, is highly related to the complexity, including scientific uncertainty, of communicating the message of a pandemic’s health risk. A second claim acknowledges integrating greater interdisciplinary expertise across expert domains between medicine/research and public health/communication that can offer possibilities for greater intelligibility of a situation at hand. Finally, a third recognizes mental models playing an important role in working through the first two points. All three are significant due to the ongoing issues surrounding pandemic health risk communication. Authors’ claims have been placed in the context of previous literature, albeit one article related to mental models is still under review (see reference 27).

Please see specific comments below.

For clarity, the sentence on lines 50-52 should be rewritten, please note:

• it has already been more than a century (1918-2022);

• there have been at least four pandemic during that time period, including 1957-58 Pandemic Flu;

• naming conventions of various pandemics are different, keep them consistent (e.g., use the disease name or the colloquial name, or utilizing both for each one), separately, if utilizing colloquial naming is undesirable across all diseases, consider extending this same care to nonhuman naming; and

• consider renaming AIDS to HIV/AIDS.

Recommend utilizing “science expert” or “research expert” as one of the four inclusive categories throughout (see below for more detail).

• Table 1 presents five sample categories (1. medicine, 2. health research, 3. medical research, 4. public health, and 5. communication) that are then integrated into four categories in lines 224-225 (1. medicine, 2. medical/health researchers (science), 3. public health, and 4. communication). The second of these expert categories is then again combined, somewhat inconsistently, in various places in the manuscript, but especially lines 569-577 (i.e., “medical and science”, “medical/health”, and “medical and research”).

Finally, if possible, but not absolutely necessary:

• Briefly clarify/define, the author’s use of “interdisciplinary” (i.e., comparing and contrasting between disciplines) in this submission.

• Note whether or not the levels of municipal, regional, and national offered any insights into how risks were perceived, contextualized, and/or communicated? (because, according to authors, there is no one-size-fits-all approach).

6. PLOS authors have the option to publish the peer review history of their article (what does this mean?). If published, this will include your full peer review and any attached files.

Reviewer #1: **Yes: **Sara Garfield

Reviewer #2: No

---

## [Author Response · Author response to Decision Letter 0]

4 Jul 2022

Please see the response to reviewer file

---

## [Decision Letter · Decision Letter 1]

27 Jul 2022

PONE-D-22-07588R1Key topics in pandemic health risk communication: A qualitative study of expert opinions and knowledgePLOS ONE

Dear Dr. Berg,

Thank you for submitting your manuscript to PLOS ONE. After careful consideration, we feel that it has merit but does not fully meet PLOS ONE’s publication criteria as it currently stands. Therefore, we invite you to submit a revised version of the manuscript that addresses the points raised during the review process.

 Note that reviewers request some minor changes in your work.

We look forward to receiving your revised manuscript.

Kind regards,

Celia Andreu-Sánchez

Academic Editor

PLOS ONE

Journal Requirements:

Reviewers' comments:

Reviewer's Responses to Questions

**Comments to the Author**

1. If the authors have adequately addressed your comments raised in a previous round of review and you feel that this manuscript is now acceptable for publication, you may indicate that here to bypass the “Comments to the Author” section, enter your conflict of interest statement in the “Confidential to Editor” section, and submit your "Accept" recommendation.

Reviewer #1: (No Response)

Reviewer #2: All comments have been addressed

2. Is the manuscript technically sound, and do the data support the conclusions?

Reviewer #1: Yes

Reviewer #2: Yes

3. Has the statistical analysis been performed appropriately and rigorously? 

Reviewer #1: N/A

Reviewer #2: N/A

4. Have the authors made all data underlying the findings in their manuscript fully available?

Reviewer #1: No

Reviewer #2: No

5. Is the manuscript presented in an intelligible fashion and written in standard English?

Reviewer #1: Yes

Reviewer #2: Yes

6. Review Comments to the Author

Reviewer #1: Thank-you for submitting your revisions.

With regards to the movement of interview length to the results section, I am not sure that where it is currently placed gives the best flow. I would suggest having a first section in the results summarising the respondent characteristics - table 1 and presenting the interview duration as part of this.

The animated figures as part of the supplementary material are a good idea. However, the figures in the main text cannot be viewed at the current size and so either they need to be simplified to allow bigger font or removed.

Reviewer #2: Only minor updates necessary:

• grammar: medical expert or expert in medicine, instead of "medicine expert" (e.g. line 281-282);

• category follow-up: change "medical/health research" to "medicine" (i.e., line 634);

• extra "the" in line 576.

7. PLOS authors have the option to publish the peer review history of their article (what does this mean?). If published, this will include your full peer review and any attached files.

Reviewer #1: No

Reviewer #2: No

---

## [Author Response · Author response to Decision Letter 1]

9 Aug 2022

R1. Reviewer #1: Thank-you for submitting your revisions.

With regards to the movement of interview length to the results section, I am not sure that where it is currently placed gives the best flow. I would suggest having a first section in the results summarising the respondent characteristics - table 1 and presenting the interview duration as part of this.

Response: We agree that this information did not fit in where it was placed. We moved this information to the heading “interviews” in the method section. In qualitative articles it is normal to place information regarding sample characteristics and interview duration in the method section. 

R1: The animated figures as part of the supplementary material are a good idea. However, the figures in the main text cannot be viewed at the current size and so either they need to be simplified to allow bigger font or removed.

Response: Thank you for notifying this issue. The Figures are enlarged and the text in Figures 1, 2 and 3 are now slightly enlarged. We choose to delete the main figure (previously named Figure 1), because the information is more readable in the zoomed-in figures. 

R2. Reviewer #2: Only minor updates necessary:

• grammar: medical expert or expert in medicine, instead of "medicine expert" (e.g. line 281-282);

Response: Corrected to expert in medicine 

R2: category follow-up: change "medical/health research" to "medicine" (i.e., line 634);

Response: Corrected to medicine

R2: • extra "the" in line 576.

Response: Corrected

---

## [Decision Letter · Decision Letter 2]

31 Aug 2022

PONE-D-22-07588R2Key topics in pandemic health risk communication: A qualitative study of expert opinions and knowledgePLOS ONE

Dear Dr. Berg,

Thank you for submitting your manuscript to PLOS ONE. After careful consideration, we feel that it has merit but does not fully meet PLOS ONE’s publication criteria as it currently stands. Therefore, we invite you to submit a revised version of the manuscript that addresses the points raised during the review process. Note that only a couple of minor comments from Reviewer 1 should be addressed now.

We look forward to receiving your revised manuscript.

Kind regards,

Celia Andreu-Sánchez

Academic Editor

PLOS ONE

Journal Requirements:

Reviewers' comments:

Reviewer's Responses to Questions

**Comments to the Author**

1. If the authors have adequately addressed your comments raised in a previous round of review and you feel that this manuscript is now acceptable for publication, you may indicate that here to bypass the “Comments to the Author” section, enter your conflict of interest statement in the “Confidential to Editor” section, and submit your "Accept" recommendation.

Reviewer #1: (No Response)

2. Is the manuscript technically sound, and do the data support the conclusions?

Reviewer #1: Yes

3. Has the statistical analysis been performed appropriately and rigorously? 

Reviewer #1: N/A

4. Have the authors made all data underlying the findings in their manuscript fully available?

Reviewer #1: No

5. Is the manuscript presented in an intelligible fashion and written in standard English?

Reviewer #1: Yes

6. Review Comments to the Author

Reviewer #1: Thank-you for your revisions. Two comments:

1. In medical journals it is standard practice to put the participant characteristics in the results section, even for qualitative papers.

2. The text in the figures is still very small.

7. PLOS authors have the option to publish the peer review history of their article (what does this mean?). If published, this will include your full peer review and any attached files.

Reviewer #1: No

---

## [Author Response · Author response to Decision Letter 2]

1 Sep 2022

Reviewer #1: Thank-you for your revisions. Two comments:

1. In medical journals it is standard practice to put the participant characteristics in the results section, even for qualitative papers

Response: We have moved participant characteristics to the result section. 

2. The text in the figures is still very small

Response: Images have been increased to 1920px wide, at 300dpi, and the text used in the images at full size is:

Font: Arial

Smallest text size: 22pt

Largest text size: 45pt

This adheres to the style guidelines for figures for PLOS one as stated in https://journals.plos.org/plosone/s/figures

---

## [Decision Letter · Decision Letter 3]

14 Sep 2022

Key topics in pandemic health risk communication: A qualitative study of expert opinions and knowledge

PONE-D-22-07588R3

Dear Dr. Røislien,

We’re pleased to inform you that your manuscript has been judged scientifically suitable for publication and will be formally accepted for publication once it meets all outstanding technical requirements.

Kind regards,

Celia Andreu-Sánchez

Academic Editor

PLOS ONE

Additional Editor Comments (optional):

Reviewers' comments:

Reviewer's Responses to Questions

**Comments to the Author**

1. If the authors have adequately addressed your comments raised in a previous round of review and you feel that this manuscript is now acceptable for publication, you may indicate that here to bypass the “Comments to the Author” section, enter your conflict of interest statement in the “Confidential to Editor” section, and submit your "Accept" recommendation.

Reviewer #1: All comments have been addressed

2. Is the manuscript technically sound, and do the data support the conclusions?

Reviewer #1: Yes

3. Has the statistical analysis been performed appropriately and rigorously? 

Reviewer #1: N/A

4. Have the authors made all data underlying the findings in their manuscript fully available?

Reviewer #1: No

5. Is the manuscript presented in an intelligible fashion and written in standard English?

Reviewer #1: Yes

6. Review Comments to the Author

Reviewer #1: Thank-you for addressing the comments. I have no further comments and recommend publication of your paper.

7. PLOS authors have the option to publish the peer review history of their article (what does this mean?). If published, this will include your full peer review and any attached files.

Reviewer #1: No

---

## [Editor Report · Acceptance letter]

22 Sep 2022

PONE-D-22-07588R3 

Key topics in pandemic health risk communication: A qualitative study of expert opinions and knowledge 

Dear Dr. Røislien:

I'm pleased to inform you that your manuscript has been deemed suitable for publication in PLOS ONE. Congratulations! Your manuscript is now with our production department. 

Kind regards, 

on behalf of

Dr. Celia Andreu-Sánchez 

Academic Editor

PLOS ONE